



# Contributors to Fluxgate Magnetic Noise in Permalloy Foils Including a Potential New Copper Alloy Regime

David M Miles[1], Richard Dvorsky[1], Kenton Greene[1], Christian Hansen[1], B. Barry Narod[2], Michael Webb[1]

[1]Department of Physics and Astronomy, University of Iowa, Iowa City, Iowa, 52242, USA
[2]Department of Earth, Ocean and Atmospheric Sciences, University of British Columbia, Vancouver, Canada

*Correspondence to*: David M Miles (david-miles@uiowa.edu)

**Abstract**

Fluxgate magnetometers provide sensitive and stable measurements of the static and low frequency vector magnetic field. Fluxgates form a magnetic field measurement by periodically saturating a ferromagnetic core and the intrinsic magnetic noise of
this material can determine the noise floor of the instrument. We present the results of an empirical experiment to understand the physical parameters that influence the intrinsic magnetic noise of fluxgate cores. We compare two permalloy alloys - the historical standard 6% molybdenum alloy and a new 28% copper alloy. We examine the influence of geometry using the historical standard 1" diameter spiral wound ring-core and a new stacked washer racetrack design. We evaluate the influence of material thickness by comparing 100 μm and 50 μm foils. Finally, we investigate heat treatments in terms of temperature and ramp rate and their role in
both grain size and magnetic noise. The results of these experiments suggest that thinner foils, potentially comprising the copper alloy, manufactured into continuous racetrack geometry washers may provide excellent performance in fluxgate sensors.

## 1    Introduction

Fluxgate magnetometers (Fornacon et al., 1999; Primdahl, 1979; Snare, 1998) provide sensitive and stable measurements of the local DC and low frequency magnetic field and have a variety of applications ranging from geophysics, space-physics and space-
weather monitoring, to marine and military sensing. The noise floor of a fluxgate is typically limited by the intrinsic magnetic noise of a ferromagnetic core that is periodically driven into magnetic saturation to modulate the local magnetic field. Much of the early work developing low-noise cores was carried out in the 1960s for military applications (e.g., Scanlon, 1966) and is not available in the scholarly literature. This work by the US Naval Surface Weapons Center (NSWC) White Oak (now a Department of Agriculture facility) and Infinetics Inc. (Scarzello et al., 2001) resulted in the standard S1000 fluxgate core, a one-inch ring-core
manufactured from a thin (3 or12 μm) foil of 6% molybdenum Permalloy with noise ranging from ~4 to ~20 pT/√Hz at 1 Hz. Müller (1998) and Musmann (2010) both describe more recent efforts achieving < 5 pT noise using a 20 μm foil for similar Permalloy. However, none of these efforts provide enough details for the manufacturing process to be understood unambiguously or to fully understand the effect of various design choices on the performance of the resulting fluxgate core.

Recent results from (Narod, 2014) established a theoretical framework for the origin of magnetic noise and magnetic hysteresis in
Permalloy foils. An initial effort (Miles et al., 2019) demonstrated and documented a preliminary manufacturing process for low noise (< 10 pT) 1" geometry ring-cores. This study expands on that work by examining the influence of foil thickness, heat treatment, geometry, and the Permalloy alloy with the goal of consistently producing lower-noise and lower-power fluxgate cores. We present the results of an initial parameter sweep comparing the traditional 6% molybdenum alloy to a new 28% copper alloy. Each alloy is tested at both 100 and 50 μm foil thicknesses with thinner foils planned to be explored as our manufacturing capability
improves. We explore the effect of the heat-treatment used to optimise magnetic noise by exploring six temperatures spanning 1000 to 1250 °C. Fluxgate cores are manufactured in two geometries – the standard spiral-wound 1" ring-core and a new



6.45x31.45 mm racetrack geometry using continuous foil washers. We explore the relationship between number of foil layers and both magnetic noise and power consumption. We examine the effect of an additional sub-Curie heat treatment at 100 °C on the intrinsic magnetic noise of the cores. We have standardised on using three layers of foil for most tests in both core geometries to

simplify manufacturing. Consequently, we are not expecting exceptional noise performance in these rings; rather we want to understand trends and to optimize the manufacturing process.

We hypothesize that magnetic noise is influenced by the grain size developed in the Permalloy in relation to the thickness of the foil. Specifically, that complete primary recrystallisation will result in lowering magnetic noise whereas secondary recrystallisation will result in higher magnetic noise. We manufactured test-coupons for every combination of material, thickness, and heat-

treatment and characterized their grain size-distribution to begin exploring this relationship.

## 2      Construction of Candidate Cores

All cores were manufactured from scratch at the University of Iowa. All permalloy alloys were melted and processed in house and all bobbins were manufactured and assembled based on in-house designs.

### 2.1      Production of Permalloy Alloy

The fluxgate cores described here were based on Permalloy manufactured at the University of Iowa. Casting small batches of Permalloy in-house allows us to more easily and cost effectively explore new metallurgy such as the copper alloy described below. High-purity source powders were combined by ratio of weight, shaken to mix, and gently packed into an Alumina crucible (Figure 1a). The crucible was placed in the process furnace under a slow purge of a reducing atmosphere of 10% hydrogen diluted in argon. The process furnace ramped from room temperature to 1550 °C at 300 °C/hour, held for one hour duration, and then returned to

room temperature at 300 °C/h. The base powders combined to form a single uniform ingot (Figure 1b) – the higher melting point molybdenum dissolving in the other liquid metals (e.g., Sene and Motta, 2013). Small cavities in the ingot were sometimes observed in the face contacting the crucible. A vacuum purge while the furnace was at 1550 °C did not significantly reduce cavity formation.

The ingots were then homogenized for 7 days at 1100 °C under a slow purge of the same reducing atmosphere. A hydraulic press was used to flatten the ingots to ~3 mm thickness (Figure 1c) after which they were machined roughly rectangular. Successive cold rolling reduced the ingot down to a 100 µm foil (Figure 1d) in ~130 passes. An additional ~10 passes further reduced a subset of the foil to 50 µm which is the present limit of our rolling mill capability.






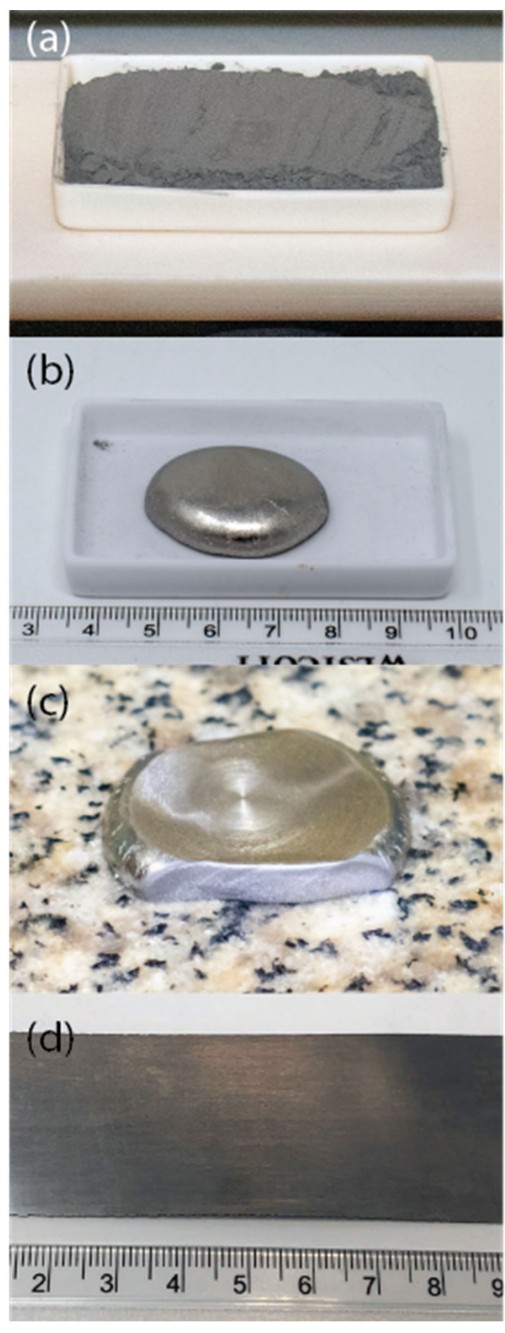

**Figure 1: Permalloy was manufactured starting from high-purity source powders that are mixed (a) and then melted together (b) via one hour at 1550 °C. The resulting ingot is flattened in a hydraulic press (c), machined rectangular, and successively cold-rolled into 100 and then 50 µm foil (d).**

### 2.2    6-81 Molybdenum Alloy [6-81 Mo]

Half of the fluxgate cores manufactured for this experiment were constructed from an alloy similar to the historical 6-81 Mo permalloy (English and Chin, 1967; Odani, 1964; Pfeifer, 1966; Pfeifer and Boll, 1969; Scanlon, 1966) combined by weight from 6% molybdenum, 81.3% nickel, and the remainder iron. The 6-81 alloy was first proposed for fluxgate use by Gordon et al. (1968)





and has been used by several groups worldwide Including Infinetics Inc., Müller et al., (1998), and Musmann (2010). The historical 6-81 alloys included ~0.5% manganese which is a common alloy additive to improve hot working, notionally by binding to sulphur contaminants. The historical 6-81 melts were hot rolled (Gordon and others, 1968), but our own melts were not, and therefore we omitted manganese.

## 2.3    28-62 Copper Alloy [28-62 Cu]

The second half of the fluxgate cores shown here were manufactured using a ferromagnetic alloy, mentioned in Narod (2014),  that we hypothesized has potential to produce high-quality fluxgate cores. This 28-62 Cu alloy was combined by weight from 28% copper, 62% nickel and the remainder iron. This ratio was derived from the theoretical framework established by Narod (2014) and prior experimental results from v. Auwers and Neumann (1935) who explored a wide range of iron-nickel-copper alloys searching for high initial permeability. V. Auwers and Neumann (1935) presented their work in a special printing from the Scientific Publications of the Siemens-Factory that was not widely circulated to the public.

The choice of 28-62 Cu for a first trial copper alloy resulted from an examination of v. Auwers and Neuman (1935) Figs 8, 11 and 13, which graph magnetic property data for a large range of iron-nickel-copper alloys, for a heat treatment process roughly similar to the ones used in our own experimental efforts. According to these old graphs, 28-62 Cu simultaneously maximizes initial magnetic permeability and minimizes magnetostriction. These are both indicators for minimum magnetic anisotropy, which according to Narod (2014) leads to the reduction of magnetostatic energy in the material and which is a direct diagnostic for improved magnetic noise. It is also the case that 28-62 Cu has expected magnetic saturation induction, $Bs$, like that of 6-81 Mo permalloy. Both theoretical calculations in Narod (2014) and $Bs$ data graphed in v. Auwers and Neuman (1935) lead to that result. For both experimental observation and theoretical reasons $Bs$ is a lead indicator for magnetic noise (Narod et al, 1985; Mussmann, 2010; Narod, 2014). 28-62 Cu's expected lower $Bs$ of 0.5-0.6 T should be advantageous.

## 2.4    Ring-core and Racetrack Variants

Two styles of fluxgate cores were manufactured to explore the impact of geometry. The classic Infinetics S1000 1" ring-core geometry was used as the standard design. We also manufactured cores in an alternative racetrack geometry (Gordon et al., 1965; Hinnrichs et al., 2000, 2001; Ripka, 1990, 1993, 2000) that combines the symmetry and closed flux path of a ring-core with the high single-axis geometric gain of a parallel rod core. The race-track geometry also allowed us to experiment with several design changes that were intended to increase reproducibility and reduce power consumptions.

## 2.5    1" Ring-Cores

The 1" ring-cores were manufactured following the description in Miles et al., (2019) as shown in Figure 2 (1a-1d). Permalloy foil was sheared into strips sufficiently long for three layers. Each strip was spot welded to the Inconel bobbin, dilute milk of magnesia was applied and dried, the foil was wound three times, and spot welded to itself aligned to the start point. The assembled ring-cores were heat-treated with variations of the standard process heat-treatment as described below. The ring-cores were then insulated with Mylar tape, toroidal drive windings of AWG 32 magnetic wire were applied, and the leads were terminated in a twisted pair.





**2.6     Racetrack Cores**

The racetrack design makes several fundamental changes beyond the gross geometry of the core assembly as shown in Figure 2 (2a-2d). The permalloy foil was cut to ~5 cm length, stacked, drilled, and secured in a tight bundle. A CNC mill was used to machine continuous 6.45x31.45 mm racetrack foil washers. No insulating coating was applied; rather the racetrack washers were placed in the furnace bare and heat-treated using variations of the standard process heat-treatment as described below, prior to

being assembled into a core. Heat-treated foils were then stacked into a non-conductive plastic bobbin (Delrin for prototypes, 30% glass filled PEEK for production) interleaved with insulating layers of Kapton film of the same geometry. A plastic lid closed the core and supports a quasi-toroidal drive of AWG 32 magnet wire. Production cores have the foil wet-set into a polymer to prevent the foil layers from moving during the magnetizing drive pulses. We are investigating foil movement as a source of long-term offset shifts.

The stacked foil washers remove the need to spot-weld the ferromagnetic element, as is done in traditional spiral-wound sensors, avoiding the heat-effected area around the weld and the associated unpredictable magnetic properties. Heat-treating the foil washers individually removes the risk of undesired welding between layers and between the foil and the standard metal Inconel bobbin that can cause unintended shorting. This process also eliminates the differential strains between Inconel and Permalloy, that invariably happens when cooling the assembly from ~1100 °C to room temperature. The race-track geometry aligns ferromagnetic mass on

one axis, potentially producing lower noise; however, the race-track geometry cannot be double-wound like a ring-core to sample two orthogonal components. The quasi-toroidal drive windings are time-consuming to apply, but the closed flux path of the racetrack should reduce stray fields and offsets error compared to traditional parallel rod sensors

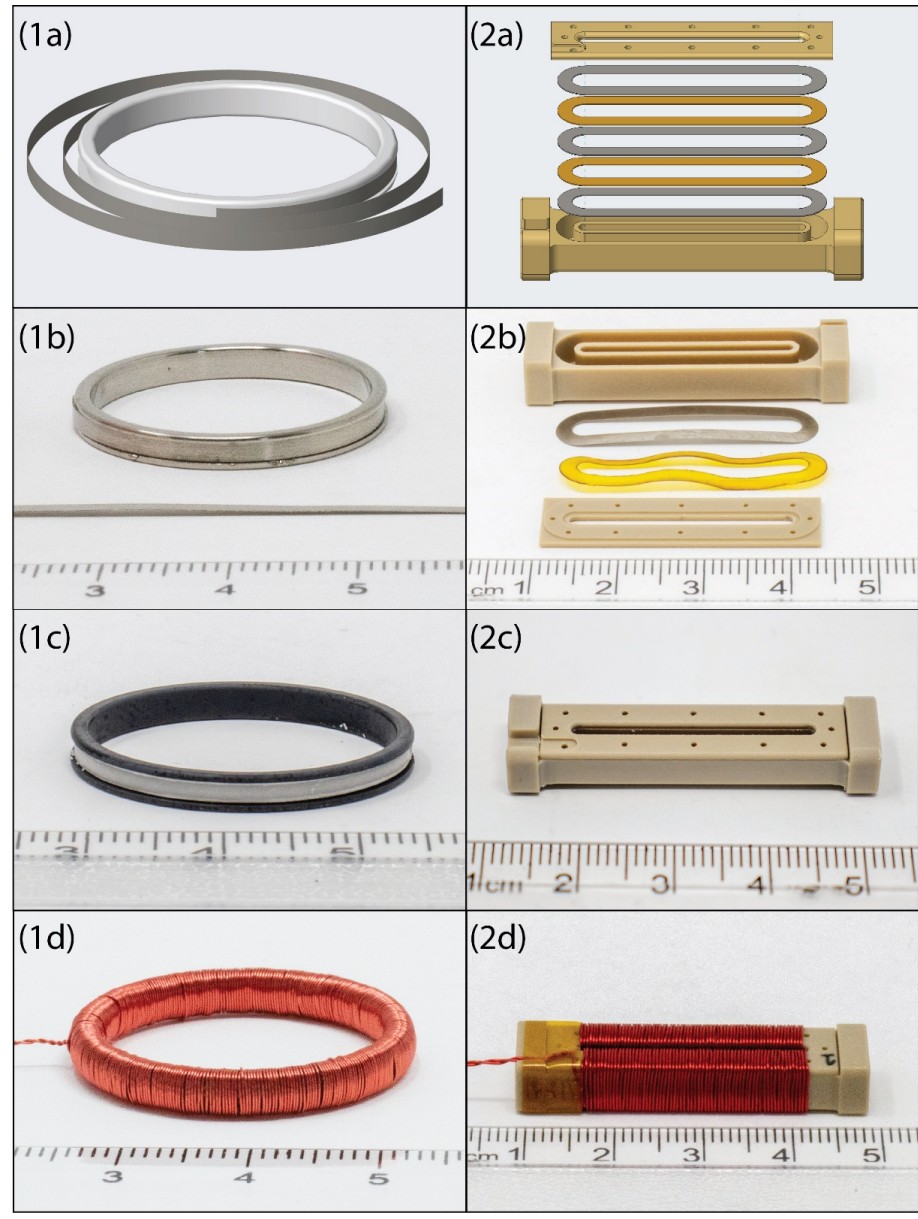

**Figure 2: Assembly of a 1" ring-core illustrated by (1a) render of principle components, (1b) photo of main components, (1c) assembled ring-core prior to application of drive-winding, and (1d) wound ring-core. (2a-d) show the same steps for the race-track geometry core.**

## 2.7 Heat Treatment

The heat treatments used here are adapted from the profile (Figure 3a) of Miles et al. (2019) which are, in turn, based loosely on a description given by Gordon et al. (1968). The heat treatment starts with the ferromagnetic material at room temperature (A) and comprises four steps: rapid heating (B) of the ferromagnetic material by insertion into the pre-heated furnace, a fixed length dwell (B – C), ramping down to the upper limit of the critical ordering range (C – D), and finally a slow ramp to room temperature (D –





E). This profile follows the theory of Narod (2014) with the goal of developing the largest possible grains in the given thickness

of the permalloy foil, enhancing primary recrystallisation through rapid heating, and suppressing secondary recrystallisation by

rapidly cooling to the disordering range.

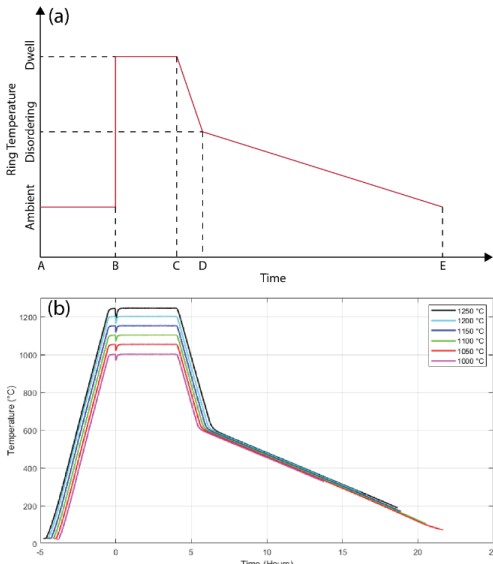

**Figure 3: (a) Illustrative ideal heat treatment - adapted from Miles et al., (2019). (b) Real-world heat-treatments. Time is normalized to the point when the material was hot loaded into the furnace indicated by the temperature dip.**

The furnace was preheated to the dwell temperature at 300 °C per hour, the fastest rate suggested by the manufacturer to avoid

cracking the Alumina work tube by thermal shock. Ramp (C-D) was completed at -300 °C per hour for the same reason. Ramp (D-

E) was completed at -35 °C per hour as used by Gordon et al. (1968). The transition point (D) was set at 600 °C at the upper limit

of the critical ordering range for 6-81 permalloy defined by Gordon et al. (1968). To explore the effect of the heat treatment, we

standardised on a four-hour dwell at temperatures ranging from 1000 to 1250 °C as shown in  Figure 3b. The transient dip in the

dwell temperature occurs when the room temperature permalloy, and the boat used to transport it, are slid into the hot zone of the

furnace. All heat treatments were completed under a continuous slow purge of a reducing atmosphere of 10% hydrogen diluted in

argon.

### 3    Core Characterisation

A common electronics package was used to drive each ring-core, measure the power consumption required, and characterize its

power spectral density noise floor. In addition, foil coupons for each alloy and thickness were included in each heat treatment so

that the grain size could be characterized optically.

### 3.1    Fluxgate Core Characterisation Setup

The various fluxgate cores were characterised using a process similar to that described in Miles et al., (2019). A common, single-

axis electronics package was used to drive and sample each fluxgate core. The 1" ring-cores were paired with a rectangular

solenoidal sense winding similar to that used in Wallis et al., (2015) while the racetrack cores were paired with a tubular solenoidal

sense coil matching their geometry (Figure 4). Fixturing allowed these sense windings to be aligned with a large solenoid, used to

generate a calibration magnetic field, mounted within a five-layer mumetal shield.



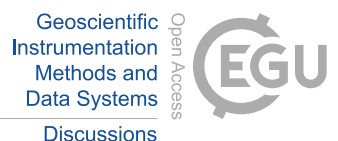

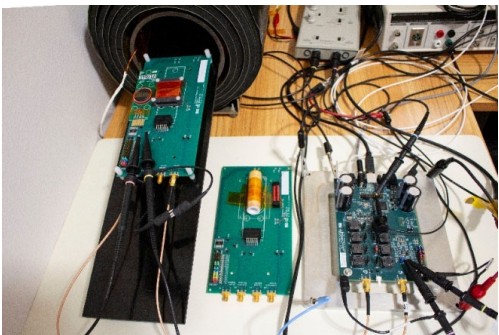

**Figure 4: Experimental setup used to characterise core noise performance: five-layer mumetal shield with embedded solenoid, test fixtures for 1" ring-core and racetrack cores, and single axis benchtop magnetometer electronics.**


A common electronics package was used to drive all cores using ±7.5V at 5.0 kHz. Both legs of the drive circuit used 1850 µH series inductors to create the resonant drive condition. For each core, the shunt capacitance was increased until symmetric current pulses were observed through the drive winding showing that minimum resonance had been achieved – then 50% additional capacitance was added.

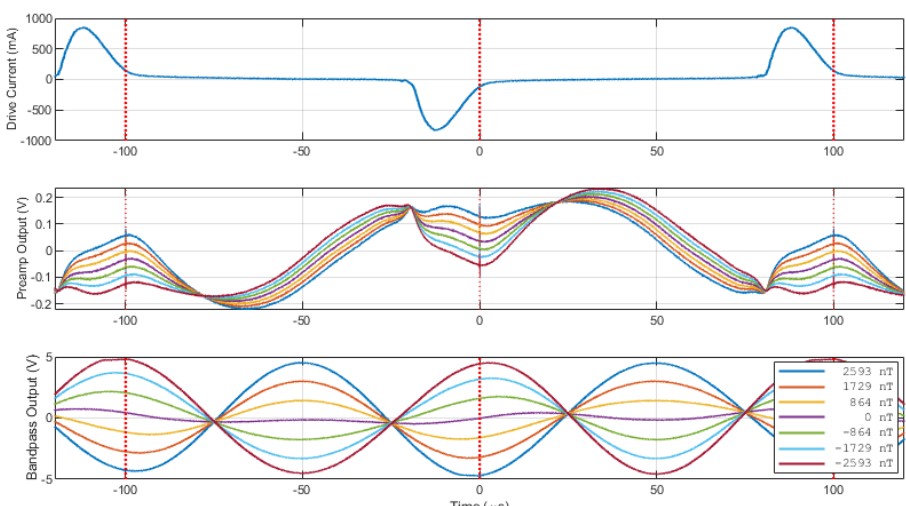


**Figure 5: (Top) Drive current periodically saturating the ring. (Middle) Output of the pre-amplifier. (Bottom) Output of bandpass filter at 2F frequency. Colours show modulation due to different applied fields.**

A common shorted-coil topology preamplifier was used for both sense windings. Several cores coupled large amounts of the 1F current waveform into the output of the sensor. To avoid electrical clipping of the pre-amplifier we standardized on a common

555-ohm feedback resistance. Although low, this accommodated all the cores shown in this experiment and provided a common comparison. A high Q bandpass filter at the 10.0 kHz 2F frequency acted as the anti-aliasing filter and provided 37 dB of gain for the fluxgate action. The filtered signal was digitized at 10.0 ksps and box-car average decimated to 100 sps before being telemetered to a computer for analysis.

### 3.2    Noise Floor Characterisation

A known-amplitude sinusoidal 1 Hz magnetic test signal was applied to the solenoid inside the five-layer mumetal shield using a signal generator and a 10 k low-tempco resistor. The phasing of the direct digitization was adjusted to maximize the amplitude of the measured test signal. A linear scaling coefficient was then adjusted until the visualization software showed the test signal with





the correct amplitude to calibrate the sensitivity of the complete single-axis magnetometer now formed around the fluxgate core under test. The 1 Hz test signal was then disabled, and 30 minutes of quiet data were taken with the core and sensor inside the

magnetic shield. Welch's method of overlapped periodograms (100 sps, 4096 point FFTs, Hann window, 75% overlap) was used to estimate the power spectral density noise floor as shown in Figure 6.

Two figures of merit were established to simplify comparing core performance. Robust linear regression (Matlab robustfit) was used to fit a linear trend to the noise floor from 0.05 to 1.0 Hz to exclude local and instrumental narrow-band noise. This trendline was evaluated at 1 Hz to produce the standard pT/√Hz at 1 Hz noise metric. We also evaluated the trendline at 0.1 Hz to reflect the

updated INTERMAGNET data requirement (Turbitt et al., 2013) for long period measurements, which has moved the noise requirement by a decade from 10 pT/√Hz at 1 Hz to 10 pT/√Hz at 0.1 Hz. The narrow bandwidth feature in Figure 6 is electronics noise related to data telemetry and is not relevant to the fluxgate core noise investigation.

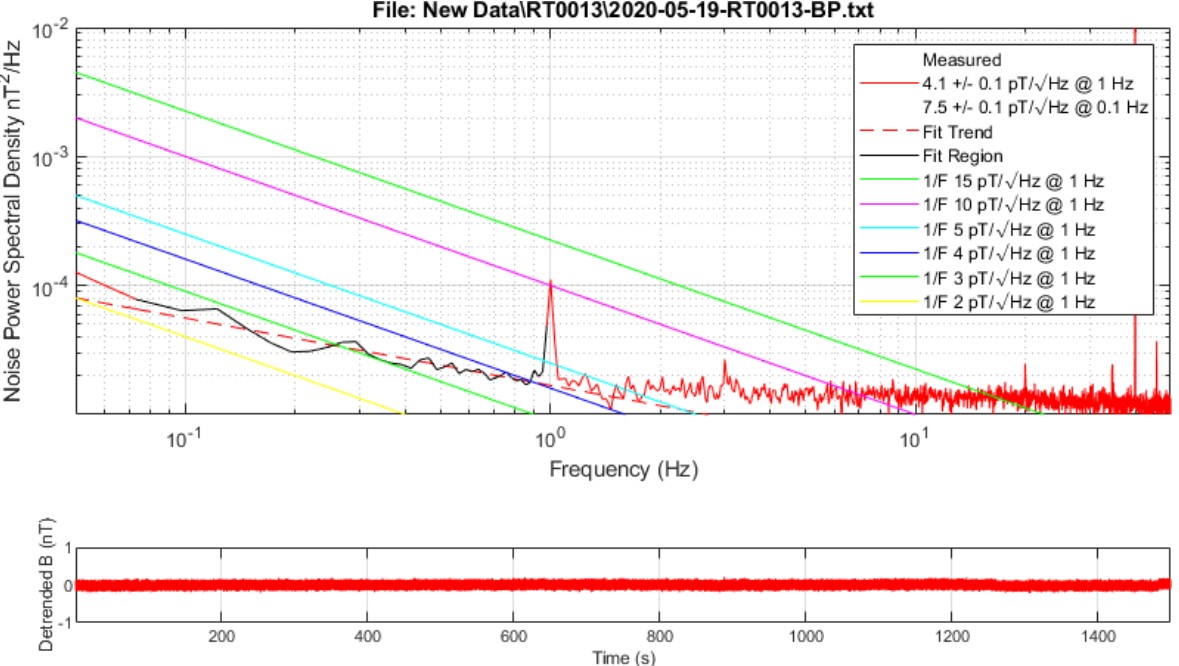

**Figure 6: Example noise floor plot for core RT0013 (6-81, 50 μm, 1150 °C dwell). Note that the noise trend below ~1 Hz**
**appears to be significantly below the expected 1/F. The narrow-band feature 1 Hz is instrumental and related to 1 sps**
**packetization and transmission over a USB interface.**

This fitting technique provides a robust, quantitative estimate of the intrinsic magnetic noise of the core despite instrumental noise at 1 Hz due to telemetry and several intermittent narrow-band magnetic noise sources in the test environment. Figure 6 shows the noise floor for a racetrack core with a measured noise of 4.1 ± 0.1 pT/√Hz at 1 Hz. It is interesting to note that the noise below 1

Hz seems to trend significantly below the historically referenced 1/F trend. This needs to be investigated further with a lower-noise preamplifier to exclude the possibility that the true trend is being masked by comparably high broadband digitizer noise.

### 3.3     Power Consumption Characterization

The drive circuit was powered by its own benchtop power supply allowing the ±0.5 mA resolution of the power required to drive each core. The drive frequency, drive voltage, and series inductance were held constant for all tests. The capacitance required to

achieve resonant drive was determined empirically as described above. Power consumption for each core was measured after each core had stabilized by operating for at least 60 seconds.





### 3.4   Grain Size Characterisation

Each heat treatment included 10 x 10 mm coupons for each combination of metallurgy and foil thickness. These coupons were used to estimate the grain size of the heat-treated material. Each coupon was photographed at 40x gain, with a graticule for scale,
using a widefield metallurgical inverted microscope under polarized visible light to emphasize the grain boundaries that had developed in the material as shown in Figure 7a.

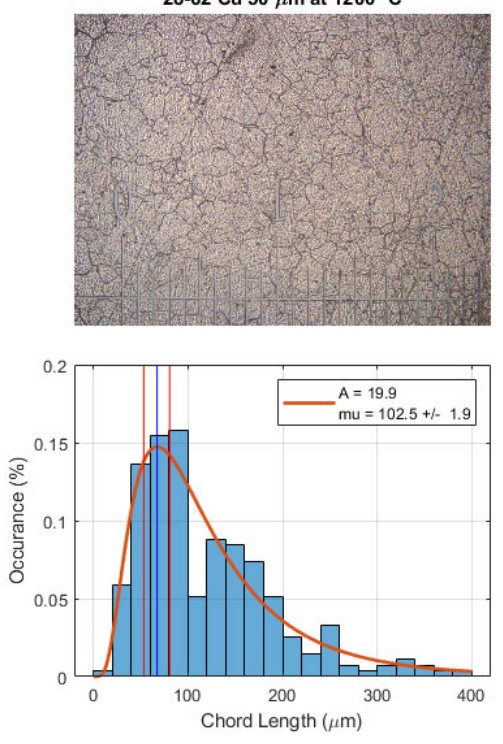

**Figure 7: (top) Photograph of 50 μm 6-81 Mo permalloy coupon heat treated at 1200 °C. Major increments in the graticule show 1 mm spacing. (bottom) Distribution of chord lengths (proxy for grain size) as determined visually using the intercept method.**

Grain size was estimated using the intercept method (Abrams, 1971) as implemented by the Matlab linecut software package by Meister (2020). Lines are drawn across each sample and chord lengths are determined by eye each time a grain boundary is crossed. The distribution of chord lengths was fit to a log-normal curve (Figure 7b) as

$$y = \frac{A}{x a_\sigma \sqrt{2\pi}} e^{\left(-\frac{(\log x - \log a_\mu)^2}{2(\log a_\sigma)^2}\right)}$$

For cross-comparison, each distribution was characterized by the mode of its probability density function (vertical blue line) plus
or minus ten percent up and down from that point on the cumulative density function (vertical red lines). Appendix A contains images (Figure A1) and cord-size distribution plots (Figure A2) of all the coupons heat-treatment as part of the optimisation effort presented here.



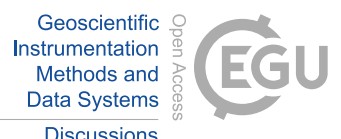

## 4     Results and Discussion

### 4.1     Core Performance

Figure 8 summarises the results of the parameter sweep of cores manufactured and tested for this experiment. The left column shows cores manufactured using the traditional 1" ring-core geometry while the right column shows the new racetrack design. The first and second rows show the measured intrinsic magnetic noise of the cores at 1 Hz and 0.1 Hz, respectively. The third row estimates the grain size observed in the foil coupon using the mode of a log-normal fit to the measured chord size distribution (Figure 11 below) plus or minus the 10% of the corresponding cumulative density function. Only a single foil coupon was

manufactured for each alloy, thickness, and heat-treatment but the data are shown in both columns for clarity. The bottom panel shows additional power required to drive each core. Blue shows the 6-81 Mo alloy while pink shows the new 28-62 Cu alloy. Finally, dashed lines indicate 100 μm foil while solid lines show 50 μm. The 100 μm 28-62 Cu racetrack core heat-treated at 1250 °C dwell coupled large amounts of 1F drive tone through the sensor saturating the preamplifier so its noise has been excluded.

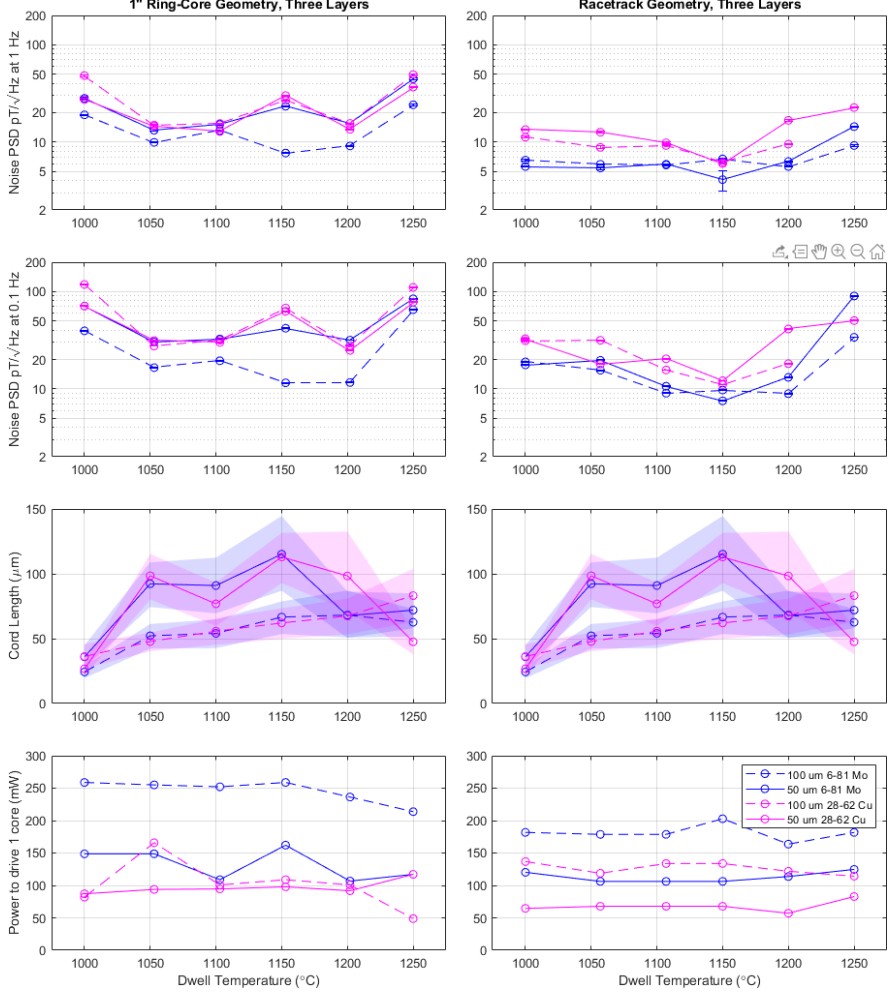

**Figure 8: Left column shows 1" ring-cores while right column shows racetrack cores. The rows show, top to bottom, noise at 1 Hz, noise at 0.1 Hz, cord length as a proxy for grain size (same data in both columns), and power required to drive each ring. All results are plotted against the dwell temperature used in their heat treatment.**



Each point in Figure 8 corresponds to a single fluxgate core and individual cores' performances are likely to vary similarly to the distribution shown in Figure 2 of Narod (2014). Therefore, we expect significant variance in the individual datapoints in addition

to the experimental uncertainty that is shown by the error bars. Overall trends in the data should be robust but individual datapoints should not be over-interpreted. Several trends are apparent.

The ring-cores' noise at 1 Hz, noise at 0.1 Hz, and power consumption varied more than the equivalent racetracks performance. We cannot isolate the origin of this variation from the current dataset but speculate it may result from inconsistency in the welding process used to attach the spiral wound foil in the 1" ring-core or due to inadvertent welding when the complete 1" ring-core

assembly is heat treated. Regardless, many of the same trends are observed in both the ring-core and racetrack data.

The 50 and 100 μm foils produced broadly similar intrinsic magnetic noise. This was not expected since, as all the cores in Figure 8 contain three layers, the 50 μm cores contain half as much ferromagnetic material as the 100 μm cores. Conversely, the lowest noise Infinetics rings known used a 12-μm foil so this may suggest 50 μm is still above the optimum foil thickness.

The measured noise as a function of heat-treatment dwell temperature, for all rings, geometries, and frequencies, are generally

concave upward with minima around 1150 °C suggesting this may be the local optimum. The 100 μm foils require more power than the equivalent 50 μm foils. In the racetrack geometry cores, the thicker foils require between 1.5 and 2 times more power. The ring-cores show an equivalent trend in 6-81 Mo while the 28-62 Cu shows a much smaller effect. More generally, the racetrack cores require less power than their ring-core counterparts – despite containing more ferromagnetic material and generally providing lower noise. Both core designs have a comparable drive winding resistance of 1.5-1.9 ohms, so it seems possible that the power

difference results from eddy current losses in the conductive Inconel x750 bobbin used in the 1" ring-core that have no equivalent in the insulating plastic bobbins used in the racetrack cores.

Optical grain size analysis of the foil coupons showed a consistent pattern of larger grain size and larger spread with increasing heat treatment temperature in the 50 μm foil consistent with accelerated grain growth at higher temperatures. The 100 μm foil cores show a maximum in grain size at ~1150 °C dwell temperature but with wider variation. We speculate that the grains in the

100 μm foil may not span the entire thickness of the foil so the optical analysis of the surface is sampling various cross sections and providing less representative values. Couderchon et al. (1989) potentially saw a similar plateau in grain size.

The 28-62 Cu material provides surprisingly respectable noise performance for a first attempt at a new alloy. Compared to the traditional 6-81 Mo alloy, 28-62 Cu provides noise performance ranging from equal to twice as noisy. However, the power required to drive the 28-62 Cu cores is roughly 30% lower than comparable 6-81 Mo cores.

The race-track cores simultaneously provide lower noise and lower power consumption than their comparable ring-core. The power advantage will be partially offset by the need for at least three race-track cores per sensor whereas only two are required for a double-wound ring-core sensor.

## 4.2 Effect of the slow Quench and Sub-Curie Heat Treatment

One heat treatment was accidently programmed to skip the second slow cool at -35 °C per hour ramp suggested by Gordon et al.

(1968). Rather, the furnace attempted to ramp down to room temperature at -300 °C per hour directly as shown in Figure 9 – the actual ramp rate being slower and low temperatures due to the thermal mass of the furnace. This provided an opportunity to investigate the effect of this long-tail in the heat treatment.

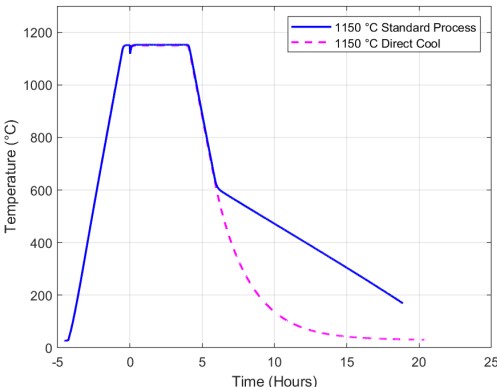

**Figure 9: Standard heat treatment with slow post-disordering cool compared to direct cooling.**

The two-resulting ring-cores (Figure 10, dashed) had an average initial noise level of ~13 pT/√Hz at 1 Hz. An equivalent set of five cores processed immediately beforehand (Figure 10, solid) with the standard long-tail heat treatment had an average initial noise of ~9 pT/√Hz at 1 Hz. Two additional heat treatments were executed to investigate whether this discrepancy was related to the cooling rate or simply time at elevated temperature below the Curie temperature. All seven rings were subjected to two rounds of an additional 100 hours at 100 °C described in Narod (2014). The initial 100 hours at 100 °C improved the noise floor of the five rings manufactured with the standard heat treatment by an average of 16% while the second 100 hours at 100 °C heat treatment provided no significant benefit. The two rings manufactured with the direct cool heat treatment showed no significant additional improvement from either 100 hour at 100 °C heat treatment.

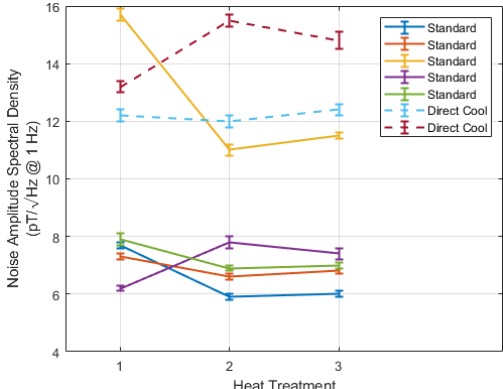

**Figure 10: Comparison of ring-cores manufactured with an initial standard heat treatment (solid) versus those with a direct cooling (dashed). Heat treatments two and 3 were each 100 hours at 100 °C.**

### 4.3 Grain Size

Figure 11 shows the distributions fitted to the chord (grain size) measurements for each alloy, foil thickness, and heat treatment. In general, hotter heat treatments increase the median chord length and broaden the distribution with 50 μm 28-62 Cu alloy showing the most consistent ordering. The trend appears to break at the 1250 °C temperature which corresponds (Figure 8) to the increasing trend in magnetic noise. We hypothesize that higher soak temperatures lead to the formation of fewer primary recrystallization initiation sites, simply because there is less time available before the dislocation energy is consumed. Thus, there are fewer grains left after primary recrystallization.

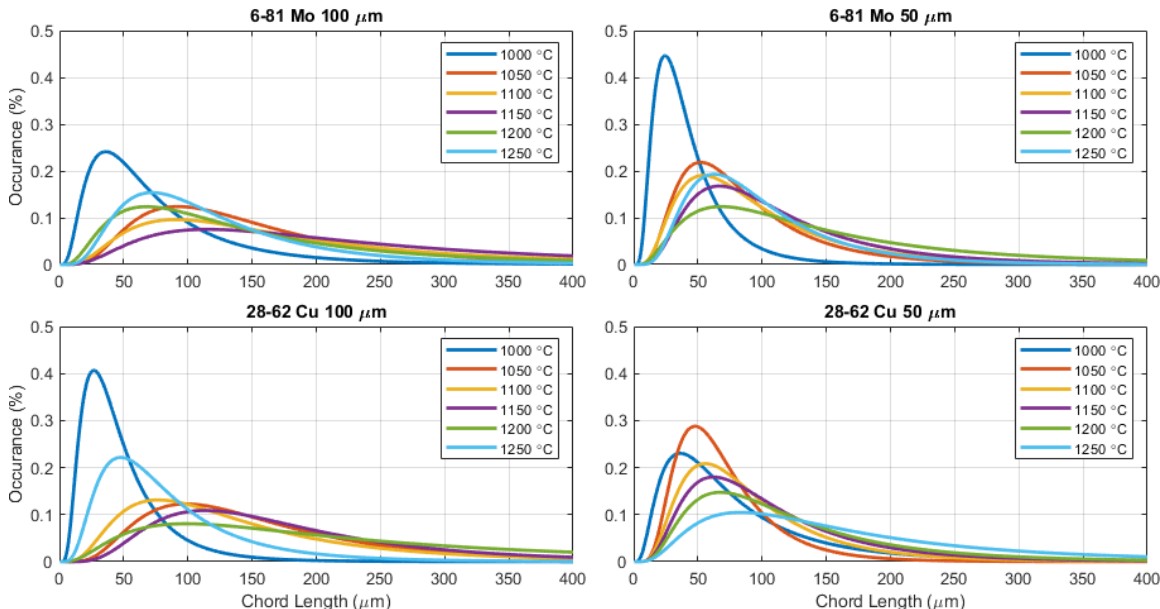

**Figure 11: Cord-size histograms for each material and thickness showing the effect of heat-treatment temperature on grain size.**

## 4.4     Effect of Number of Layers

A batch of four racetrack cores were manufactured to explore the relationship between magnetic noise, power consumption, and the number of foil layers. The four cores contained 1, 2, 6, and 9 layers of 50 µm 28-62 Cu alloy and had a common heat treatment. Figure 12 shows that increasing the number of foil layers decreases the magnetic noise at 0.1 and 1.0 Hz with the effect diminishing

between 6 and 9 layers. Power consumption increases linearly with the number of layers.

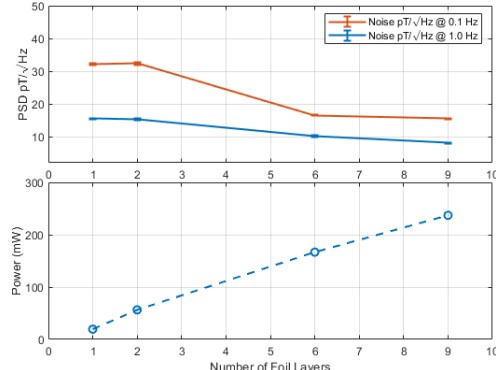

**Figure 12: Magnetic noise and power consumption as a function of number of foil layers.**

## 4.5     Noise Distribution

We manufactured twenty notionally identical racetrack cores to test the noise variability of the manufacturing process. All cores

were three layers of 50 µm 28-62 µm foil heat-treated with a dwell temperature of 1150 °C. Figure 13 shows a histogram of the noise floor distributions showing peaks at ~16 pT/√Hz at 0.1 Hz and 9 pT/√Hz at 1 Hz.

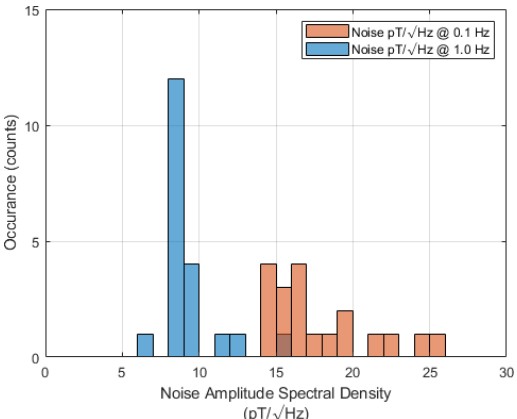

**Figure 13: Histogram of noise distribution for 20 cores**

### 5 Conclusions

The race-track washer design seems to offer several advantages over the traditional 1" spiral wound ringcore: more consistent yield in noise performance, significantly fewer high-noise outliers and, lower noise performance at equal or lower power consumption per core. These advantages will be partially offset by the need for at least one core per measurement axis (three cores per sensor) whereas ring-cores can be double wound (two cores per sensor). In general, the 50 μm foils outperform the 100 μm foils in terms of noise per drive power.

In this paper we have introduced a novel, low-noise and low-power nickel alloy we have designated 28-62 Cu-permalloy. It comprises, by weight, 28% Cu, 62% Ni, Fe balance. This alloy is suggested by data published in v.Auwers and Neumann (1935), and theory in Narod (2014). In 1935 high Cu-content permalloys were investigated for more typical, high saturation induction applications, and fell out of favour. We believe that we are the first investigators since then to revisit Cu-permalloys, and specifically with regard to fluxgate sensors where lower saturation induction is advantageous.

The 6-81 Mo alloy generally produces lower noise than the 28-62 Cu alloy, all other variables held equal. However, the 28-62 requires significantly lower power. The consistent grain size and evolution shown by the 50 μm coupon suggests that the grains developed in the foil may be spanning the entire thickness of the foil whereas in the 100 μm foils the grains may not penetrate the complete thickness. The 1150 °C dwell temperature appears to provide the lowest noise for all alloy and foil thickness combinations. Directly cooling at 300 °C / h down to room temperature appears to significantly degrade noise performance in a

way that cannot be improved by subsequent sub-curie heat treatment. However, the 100 hours at 100 °C sub-curie secondary heat-treatment offers a significant improvement in cores manufactured using the standard process suggesting it be added to the standard process or the second cooling rate be decreased. Additional layers of foil appear to reduce the magnetic noise until ~6-9 layers. Each additional foil layer causes a linear increase in the power consumption.

### 6 Future Work

These results suggest that thinner foils may yield superior noise and power performance which is consistent with 12.5 μm foil used in the best historical Infinetics cores. We are currently developing improved rolling mill capacity with the intention of investigating 25 and 12.5 μm foils. The 28-62 Cu alloy is promising in terms of noise relative to drive power – particularly for a first attempt in a new alloy regime. We intend to explore nearby copper alloy ratios based on the parameter space defined by Auwers and Neumann



( 1935). In general, race-track geometry core containing more layers of thinner copper alloy foil seems like a promising path to

consistently manufacturable, low-noise and low-power fluxgate cores.





**Appendix A: Grain Photos and Cord-size Distributions**

| | 6-81 Mo 100 µm | 6-81 Mo 50 µm | 28-62 Cu 100 µm | 28-62 Cu 50 µm |
|---|---|---|---|---|
| 1000 °C | | | | |
| 1050 °C | | | | |
| 1100 °C | | | | |
| 1150 °C | | | | |
| 1200 °C | | | | |
| 1250 °C | | | | |

**Figure A1: Photographs of each coupon showing grain size. Major ticks in the graticules show 1 mm spacing.**








**Figure A2: Cord-size distribution for each foil coupon.**

## 7    Code and Data Availability

Data and source code used in the creation of this paper can be accessed by contacting the authors.



**8    Author Contributions**

D. M. Miles wrote the manuscript with contributions from all authors. R. Dvorsky led the design of the heat-treatment furnace used to manufacture the cores described here. K. Greene assisted with the characterisation of the experimental fluxgate cores. C. Hansen led the development and manufacture of the race-track core design. B. B. Narod provided the underlying theory of the new copper alloy and the scientific interpretation of the relation between grain structure and magnetic noise. M. Webb assisted with 350 core manufacturing and imaged the foil coupons.

**9    Competing Interests**

B. B. Narod operated Narod Geophysics Ltd., which manufactured fluxgate magnetometers until the company ceased production operation in 2008. D. M. Miles and B. B. Narod hold provisional US patent 63/164,045 related to the use of the described copper alloy regime for magnetic field instruments and magnetic shielding.

**10    Acknowledgements**

This material is based upon work supported by the National Aeronautics and Space Administration under Grant No. 80NSSC19K0491 issued through the Science Mission Directorate and Contract No. 80GSFC18C0008 administered by Goddard Space Flight Center.

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

Narod, B. B.: The origin of noise and magnetic hysteresis in crystalline permalloy ring-core fluxgate sensors, Geosci. Instrum.
Methods Data Syst., 3, 201, 2014.

Odani, Y.: Magnetic Properties of Cube-Textured 6–81.3 Mo—Permalloy, J. Appl. Phys., 35, 865–866, 1964.

Pfeifer, F.: Zum Verstandnis der magnetischen Eigenschaften technischer Permalloylegierungen, Z. Met., 57, 295, 1966.

Pfeifer, F. and Boll, R.: New soft magnetic alloys for applications in modern electrotechnics and electronics, IEEE Trans. Magn.,
5, 365–370, 1969.

Primdahl, F.: The fluxgate magnetometer, J. Phys. [E], 12, 241, 1979.

Ripka, P.: Improved fluxgate for compasses and position sensors, J. Magn. Magn. Mater., 83, 543–544,
https://doi.org/10.1016/0304-8853(90)90618-Z, 1990.

Ripka, P.: Race-track fluxgate sensors, Sens. Actuators Phys., 37–38, 417–421, https://doi.org/10.1016/0924-4247(93)80071-N,
395   1993.

Ripka, P.: Race-track fluxgate with adjustable feedthrough, Sens. Actuators Phys., 85, 227–231, https://doi.org/10.1016/S0924-
4247(00)00394-0, 2000.

Scanlon, W. W.: Solid state research of the Applied Physics Department for the year 1965, NAVAL ORDNANCE LAB WHITE
OAK MD, 1966.

Scarzello, J. F., Holmes, J. J., and O'keefe, E. C.: Integrating fluxgate magnetometer, 2001.

Sene, F. F. and Motta, C. C.: Synthesis and characterization of Ni-Mo filler brazing alloy for Mo-W joining for microwave tube
technology, Mater. Res., 16, 417–423, https://doi.org/10.1590/S1516-14392013005000019, 2013.

Snare, R. C.: A History of Vector Magnetometry in Space, in: Measurement Techniques in Space Plasmas Fields, edited by: Pfaff,
R. F., Borovsky, J. E., and Young, D. T., American Geophysical Union, 101–114, https://doi.org/10.1002/9781118664391.ch12,
405   1998.

Turbitt, C., Matzka, J., Rasson, J., St-Louis, B., and Stewart, D.: An instrument performance and data quality standard for
intermagnet one-second data exchange, in: Proceedings of the XVth IAGA Workshop on Geomagnetic Observatory Instruments,
Data Acquisition, and Processing, XVth IAGA Workshop on Geomagnetic Observatory Instruments and Data Processing, Cadiz,
Spain, 186–188, 2013.

Wallis, D. D., Miles, D. M., Narod, B. B., Bennest, J. R., Murphy, K. R., Mann, I. R., and Yau, A. W.: The CASSIOPE/e-POP
Magnetic Field Instrument (MGF), Space Sci. Rev., 189, 27–39, https://doi.org/10.1007/s11214-014-0105-z, 2015.