# Peer review of "Contributors to Fluxgate Magnetic Noise in Permalloy Foils Including a Potential New Copper Alloy Regime"

_Geoscientific Instrumentation, Methods and Data Systems, 2021_

## Author Comment (AC1)

Response to Interactive comment RC1 on gi-2021-30 - "Contributors to Fluxgate Magnetic Noise in Permalloy Foils Including a Potential New Copper Alloy Regime" by David M. Miles et al. by Dave Sheppard on 30 Nov 2021

*We thank the referee for the constructive comments which we have incorporated into the manuscript. Dave Sheppard raised an important issue, which we address below. Referee comments are in plain text our responses in italics and any content added to or changed in the manuscript are in "quoted italics"*

This work is important and relevant to the field of magnetometry. The new copper alloy is novel and builds upon work done many decades ago. The direct comparison with the research standard 1" ring core permalloy sensors is useful in evaluation of the materials. The construction and evaluation of the material is well described in this paper. That is unique in that work done several decades ago on this topic was either not well described or not published in unclassified documentation. This is a good reference for fluxgate materials going forward.

The paper is clear and well written. The plots and diagrams are appropriate and well done. The references are good.

One small comment: On line 126, we read "e quasi-toroidal drive windings are time-consuming to apply, but the closed flux path of the racetrack should reduce stray fields and offsets error compared to traditional parallel rod sensors" However, there are neither measurements nor references used to support this statement. Additionally, the racetrack is compared to parallel rods here, but in the rest of the paper, the racetrack topology is compared to ring cores. This section is potentially improved by citing offset measurements, providing reference(s), and/or comparing the racetrack versus ring core topologies, as is done in the remainder of the document, in place of the parallel rods.

*Our intention here was to explain one of the reasons why the racetrack geometry was selected for experimentation rather than to assert results beyond the scope of this manuscript. We have rephrased this section to make this clear and added relevant references.*

*Change made. Text now reads:*

*"The quasi-toroidal drive windings are time-consuming to apply; however, some literature (e.g. Janosek, 2017) suggests that the closed flux path of the racetrack may reduce magnetic noise, stray fields, and offsets error compared to traditional parallel rod sensors by avoiding the presence of an un-saturated or weakly saturated end region (e.g., Moldovanu et al., 2000) and this will be examined in future work."*

This is a good work and I look forward to following this research in the future. It is both important and relevant to the field of fluxgate magnetometry.

---

## Author Comment (AC2)

Response to Interactive comment RC2 on gi-2021-30 - "Contributors to Fluxgate Magnetic Noise in Permalloy Foils Including a Potential New Copper Alloy Regime" by David M. Miles et al. by Anonymous Referee #2 on 12 Dec 2021

*We thank the referee for the constructive comments which we have incorporated into the manuscript. Anonymous Referee #2 raised several important questions and issues, which we address below. Referee comments are in plain text our responses in italics and any content added to or changed in the manuscript are in "quoted italics"*

Excellent work, which demonstrates, including technological details of core production for fluxgate magnetometers. The noise of fluxgate sensors made using cores of fundamentally different alloys (6-81 Mo and 28-62 Cu) and with different geometries (ring and racetrack) is studied. The authors describe in detail the subtleties of heat treatment of alloys, which significantly improves the value of magnetic noise for both used alloys. It has been shown that fluxgate sensors with racetrack core have lower noise values ‹â ‹and lower power consumption compared to ring cores. The authors showed that the alloy 28-62 Cu has slightly higher noise values ‹â ‹compared to the classical alloy 6-81 Mo. The advantage of the 28-62 Cu alloy is lower power consumption. The authors claim that this alloy is promising and in the future, after its improvement, it may compete with 6-81 Mo permalloy.

In general, the paper presents a large number of important results. This work will be useful for technologists and scientists working in the field of magnetic sensor development.

I have a few comments and suggestions.

In paragraph 2.6 Racetrack Cores (line 110 et seq.) It would be desirable to indicate what was the width of the foil in the ring core (1.57 mm, as in https://doi.org/10.5194/gi-2019-15)? Was it the same or comparable to the width of the straight foil segments in the racetrack core (6.45 mm minus the width of the hole (cut oval))?

*Change made. The foil in ring core was indeed 1.57 mm as in previous publications. The foil width in the race track geometry was 1.70 mm as shown in the new Figure 2.*

*Section 2.5 now reads:*

*"Permalloy foil was sheared into 1.57 mm wide strips …"*

*Section 2.6 now reads:*

*"A CNC mill was used to machine continuous 6.45 mm wide by 31.45 mm long racetrack foil washers (Figure 2). The track width was machined to 1.70mm. The gap being sized to comfortable accommodate hand-winding of a toroidal drive winding."*

[Figure]

**Figure 1: Dimensions of the machined ferromagnetic foil used in the racetrack core.**

In line 116 the phrase "A plastic lid closed the core and supports a quasi-toroidal drive of AWG 32 magnet wire" it would be desirable not to use the combination "magnet wire". Better: "A plastic lid closed the core and supports a quasi-toroidal drive of AWG 32 wire, which induces a magnetic flux."

Line 305 has a mistake: All cores were three layers of 50 µm 28-62 Cu ("µm" -please, remove) foil heat-treated with a dwell temperature of 1150 °C.

*Change made.*

Figure 13 shows a dark brown bar inw the range of 15÷16 pT/√Hz. However, there is no explanation for this color in the figure.

*Change made. The 'brown bar' resulted from the overlap of the 0.1 Hz and 1.0 Hz spectra. The figure has been regenerated with semi-transparent bars and hash lines for clarity. The caption has been amended accordingly.*

*Figure is now:*

[Figure]

**Figure 2: Histograms of noise distribution for 20 cores. Bin 16 pT/√Hz includes cores from both histograms**

It would be desirable to supplement the work with studies of the integral magnetic properties of cores, in particular, as was done for permalloy ring cores in the work of David M Miles and co-authors

(https://doi.org/10.5194/gi-2019-15). For example, to build the dependence of the sensor noise on the maximum magnetic permeability (or/end core loss) of cores made of different alloys.

*Change made. The test of the alloy 28-62 Cu is only our first trial of a copper permalloy. Based on these results we have begun an extensive survey of alloys of nearby compositions, guided by magnetic properties data presented in v. Auwers and Neumann (1935). We plan to cover a composition range from 28% to 51% copper, and also covering the zero magnetostriction range. We will be examining DC resistivity, coercivity, saturation moment, initial and maximum differential permeability, magnetostriction, Curie temperature, grain size, and grain growth fabrics with some of these properties examined at temperatures from room temperature through to Curie temperature.*

*Alloy 28-62 Cu was chosen as the first trial due to it having magnetic properties similar to those of 6-81 Mo in the molybdenum permalloy composition range (Pfeifer and Boll, 1969). 28-62 Cu has approximately zero magnetostriction and has minimum magnetocrystalline anisotropy as evidenced by its local maximum initial permeability and minimum coercivity, with all these properties achieved by identical heat treatment.*

*Text discussing this has been added to Section 6.*

My remarks and wishes, of course, do not reduce the value and novelty of the results of the research presented by the authors.